# Calcitonin Gene-Related Peptide mRNA Synthesis in Trigeminal Ganglion Neurons after Cortical Spreading Depolarization

**DOI:** 10.3390/ijms241411578

**Published:** 2023-07-18

**Authors:** Mamoru Shibata, Satoshi Kitagawa, Miyuki Unekawa, Tsubasa Takizawa, Jin Nakahara

**Affiliations:** 1Department of Neurology, Keio University School of Medicine, Tokyo 160-8582, Japan; sts.ktgw@gmail.com (S.K.); unekawa.m@z5.keio.jp (M.U.); tsubasa.takizawa@z5.keio.jp (T.T.); nakahara@a6.keio.jp (J.N.); 2Department of Neurology, Tokyo Dental College Ichikawa General Hospital, Chiba 272-8513, Japan

**Keywords:** migraine, cortical spreading depolarization (CSD), calcitonin gene-related peptide (CGRP), trigeminal ganglion, in situ hybridization

## Abstract

Migraine is a debilitating neurovascular disorder characterized by recurrent headache attacks of moderate to severe intensity. Calcitonin gene-related peptide (GGRP), which is abundantly expressed in trigeminal ganglion (TG) neurons, plays a crucial role in migraine pathogenesis. Cortical spreading depolarization (CSD), the biological correlate of migraine aura, activates the trigeminovascular system. In the present study, we investigated *CGRP* mRNA expression in TG neurons in a CSD-based mouse migraine model. Our in situ hybridization analysis showed that *CGRP* mRNA expression was observed in smaller-sized neuronal populations. CSD did not significantly change the density of *CGRP* mRNA-synthesizing neurons in the ipsilateral TG. However, the cell sizes of *CGRP* mRNA-synthesizing TG neurons were significantly larger in the 48 h and 72 h post-CSD groups than in the control group. The proportions of *CGRP* mRNA-synthesizing TG neurons bearing cell diameters less than 14 μm became significantly less at several time points after CSD. In contrast, we found significantly greater proportions of *CGRP* mRNA-synthesizing TG neurons bearing cell diameters of 14–18 μm at 24 h, 48, and 72 h post-CSD. We deduce that the CSD-induced upward cell size shift in *CGRP* mRNA-synthesizing TG neurons might be causative of greater disease activity and/or less responsiveness to CGRP-based therapy.

## 1. Introduction

Migraine adversely affects individuals and society as the second leading cause of years lived with disability worldwide [1]. Migraine is characterized by recurrent headache attacks that are preceded by transient neurological symptoms referred to as aura in some cases [2]. Migraine aura emerges as visual scintillations and scotoma in most cases and, less often, as hemisensory symptoms or dysphasia. In typical cases, these neurological symptoms last 5–60 min, and headache ensues. This temporal sequence of migraine attacks has been attracting much interest from headache researchers. Migraine aura is caused by cortical spreading depolarization (CSD), a concentrically propagating wave of abrupt and sustained near-complete breakdown of transmembrane ion gradient and mass depolarization in the brain tissue [3,4,5,6,7,8,9,10]. CSD has been shown to activate the trigeminal system [11,12,13], indicating that CSD may be responsible for migraine headache as well as migraine aura. The putative mechanisms whereby CSD induces the trigeminal system include meningeal neurogenic inflammation, parenchymal neuroinflammation, and cortical metabolic changes [14]. CSD is widely used to produce a migraine model in animal studies [4,15,16,17,18]. It is known that CSD causes sustained electrical activation of rat trigeminal ganglion (TG) neurons on the ipsilateral side [11]. This finding was reinforced by the subsequent observation that neuronal activation was initiated after CSD in laminae I-II of the trigeminal nucleus caudalis [12]. Traditionally, CSD has been induced mainly by KCl application or pinprick stimulation on the cortical surface, which necessitates invasive surgical procedures, including craniotomy. Consequently, there is a concern that such invasiveness may render the interpretation of data difficult, especially in pain/headache-related studies.

Calcitonin gene-related peptide (CGRP) is expressed in TG neurons that provide meningeal sensory innervation, especially in the perivascular area [19,20]. Peripheral action of CGRP is likely to play a pivotal role in the generation of migraine headache because CGRP-targeted monoclonal antibodies, which do not readily cross the blood–brain barrier, confer potent migraine prevention [21]. Trigeminal activation is known to induce CGRP release from the meningeal trigeminal fibers [22,23,24]. Several lines of evidence indicate that CGRP induces the sensitization of nociceptors [25,26,27,28,29,30]. The sensitization of trigeminal nociceptors renders normally innocuous mechanical stimulation from vascular pulsation noxious stimuli, which is ultimately perceived as pulsating headache in the brain [20]. For the assessment of trigeminal activation relevant to migraine pathophysiology, it is crucial to investigate the status of de novo CGRP synthesis in TG neurons.

In the present study, we explore the effects of CSD on *CGRP* mRNA expression in TG neurons by employing in situ hybridization (ISH) of TG tissue and a relatively noninvasive CSD induction method that requires neither craniotomy nor electrode installation into the brain parenchyma.

## 2. Results

### 2.1. CSD Induction

Our experimental setting for CSD induction and recording is depicted in Figure 1A. The occurrence of five CSD episodes in the left cerebral hemisphere was confirmed by observing regional cerebral blood flow (rCBF) changes detectable by laser Doppler flowmetry (Figure 1B). The TG tissue was successfully sampled from CSD-subjected mice at 6, 24, 48, and 72 h after CSD induction (*n* = 3 at each time point). In addition, we used untreated mice as controls (*n* = 2). Hence, the following experimental groups were studied in the present study: the control, 6 h post-CSD, 24 h post-CSD, 48 h post-CSD, and 72 h post-CSD groups.

### 2.2. ISH for Mouse CGRP mRNA in TG Tissue

We initially designed four different ISH probes (Calca-1, Calca-2, Calca-3, and Calca-4) for mouse *CGRP* mRNA. In preliminary experiments, we visualized *CGRP* mRNA expression in mouse embryonic day 18.5 thyroid tissue using these probes, which revealed that the Calca-4 probe achieved the greatest staining performance. The ISH of mouse TG tissue using the Calca-4 anti-sense probe identified *CGRP* mRNA expression mainly in small- to medium-sized neurons (Figure 2A). On the other hand, there was no significant staining in ISH analysis using the Calca-4 sense probe (Figure 2B). From these results, we used the Calca-4 probe for further ISH studies.

### 2.3. Density of CGRP mRNA-Synthesizing TG Neurons after CSD

We examined the density of *CGRP* mRNA-synthesizing TG neurons on the CSD side in the control and CSD-subjected mice. Six tissue sections were studied in each experimental group. Representative TG tissue photographs of ISH for *CGRP* mRNA are shown in Figure 3. The density of *CGRP* mRNA-synthesizing TG neurons in control mice was 214.4 ± 102.4 (mean ± SD) cells/mm^2^, which served as the reference value. CSD did not exert any significant effect on the density of *CGRP* mRNA-synthesizing TG neurons (181.4 ± 75.4 cells/mm^2^ at 6 h post-CSD, 209.3 ± 75.4 cells/mm^2^ at 24 h post-CSD, 214.4 ± 90.6 cells/mm^2^ at 48 h post-CSD, and 186.3 47.0 cells/mm^2^ at 72 h post-CSD; *p* = 0.9373, Kruskal–Wallis test followed by Dunn’s multiple comparison test; see Figure 4).

### 2.4. Cell Size Changes in CGRP mRNA-Synthesizing TG Neurons after CSD

The cell size of *CGRP* mRNA-synthesizing TG neurons in the control mice was 14.9 ± 6.0 (mean ± SD) μm (*n* = 515). One-way analysis of variance (ANOVA) detected a significant effect of CSD on the cell size of *CGRP* mRNA-synthesizing TG neurons ipsilateral to CSD induction (F_(4, 2862)_ = 8.838, *p* < 0.0001). As shown in Figure 5, the cell sizes of *CGRP* mRNA-synthesizing TG neurons in the 48 h post-CSD and 72 h post-CSD groups were significantly greater as compared to the control group (mean differences: 0.89 [95% CI: 0.00–1.77] μm, 48 h post-CSD group [*n* = 602] vs. control group (*n* = 515), *p* = 0.0492; 1.48 [95% CI: 0.61–2.35] μm, 72 h post-CSD group (*n* = 648) vs. control group (*n* = 515), *p* = 0.0001; Dunnett’s multiple comparison test). We performed ISH for *β-actin* mRNA using TG tissue sections (Figure 6). There were no significant differences in the cell sizes of *β-actin* mRNA-synthesizing TG neurons on the CSD side between the control and 72 h post-CSD groups (19.29 ± 5.14 (mean ± SD) μm in the control group vs. 19.59 ± 5.40 (mean ± SD) μm in the 72 h post-CSD group, *p* = 0.5699, unpaired Student’s *t*-test, *n* = 200 in each group). In the control group, *CGRP* mRNA-synthesizing TG neurons were significantly smaller than *β-actin* mRNA-synthesizing TG neurons (mean difference: −4.37 [95% CI: −5.32 to −3.43] μm, *p* = 0.0105, unpaired Student’s *t*-test). On the contralateral (right) side, the cell size of *CGRP* mRNA-synthesizing TG neurons in the control mice was 15.2 ± 5.6 (mean ± SD) μm (*n* = 520). The cell sizes of *CGRP* mRNA-synthesizing TG neurons on the contralateral non-CSD side were 15.5 ± 5.4 μm (*n* = 583) at 6 h, 15.0 ± 5.5 μm (*n* = 588) at 24 h, 15.4 ± 5.4 μm (*n* = 457) at 48 h, and 15.8 ± 5.0 μm (*n* = 648) at 72 h post-CSD. We did not detect a significant effect of CSD on the cell size of *CGRP* mRNA-synthesizing TG neurons (F_(4, 2791)_ = 2.242, *p* = 0.0621).

### 2.5. Changes in Cell Size Distributions of CGRP mRNA-Synthesizing TG Neurons after CSD

We next explored the effect of CSD on the cell size distribution of *CGRP* mRNA-synthesizing TG neurons. The proportions of *CGRP* mRNA-synthesizing TG neurons bearing cell diameters less than 10 μm were significantly reduced in the 24 h post-CSD and 72 h post-CSD groups as compared to the control group (*p* = 0.0067, 24 h post-CSD group vs. control group; *p* = 0.0023, 72 h post-CSD group vs. control group, chi-square test, Figure 7). The proportions of *CGRP* mRNA-synthesizing TG neurons bearing cell diameters of 10–14 μm were significantly less in the 48 h post-CSD and 72 h post-CSD groups than in the control group (*p* = 0.0054 in the 24 h post-CSD group and *p* = 0.0012 in the 72 h post-CSD group, chi-square test, Figure 7). On the other hand, we found significantly greater proportions of *CGRP* mRNA-synthesizing TG neurons bearing cell diameters of 14–18 μm in the 24 h post-CSD, 48 h post-CSD, and 72 h post-CSD groups than in the control group (*p* = 0.001 in the 24 h post-CSD group, *p* = 0.0148 in the 48 h post-CSD group, and *p* = 0.0009 in the 72 h post-CSD group, chi-square test, Figure 7). The proportion of *CGRP* mRNA-synthesizing TG neurons bearing cell diameters of 18–22 μm was significantly less in the 6 h post-CSD group than in the control group (*p* = 0.0459, chi-square test, Figure 7).

## 3. Discussion

The present study showed that CSD induced in the left cerebral hemisphere did not cause any significant change in the cell density of *CGRP* mRNA-synthesizing TG neurons on the ipsilateral side for up to 72 h. However, *CGRP* mRNA-synthesizing TG neurons on the CSD side in the 48 h and 72 h post-CSD groups were significantly larger than those in the control group. This phenomenon was not attributable to cell swelling because we did not find any change in the cell size of *β-actin* mRNA-synthesizing TG neurons in the sections prepared from the same TG tissue. Such a significant cell size change was not observed on the contralateral side. Of note, our cell size distribution analysis revealed that CSD induced a shift in the TG neurons with the greatest *CGRP* mRNA production toward a larger population on the ipsilateral side. Plus, it merits noting that we adopted a relatively noninvasive CSD induction method in the present study.

Although CGRP is a bona fide therapeutic target for migraine, it remains largely unknown whether CSD may change the expression status of this neuropeptide in the nervous system. In rat CSD experiments, RT-PCR assays demonstrated increased *CGRP* mRNA expression in the ipsilateral cerebral cortex [31] and the amygdala [32,33]. Moreover, CSD was found to increase the amount of CGRP at the peptide level in the ipsilateral cerebral cortex [31]. Intriguingly, the CGRP receptor antagonist, olcegepant, reduced the occurrence of CSD [34], which suggests that a vicious cycle can be formed between CSD and CGRP release. Much less is known about the effect of CSD on CGRP production and release in TG tissue. Recurrent CSD over 90 min led to an increase in CGRP-immunoreactive neurons in TG tissue ipsilateral to CSD induction [35]. In accord, Yisarakun et al. [36] reported that the induction of CSD significantly increased the average percentage of total CGRP-immunoreactive neurons as compared to the control groups in TG tissue ipsilateral to CSD induction at 2 h in the absence of *CGRP* mRNA upregulation. However, the information about the effect of CSD on CGRP expression in TG tissue during the post-CSD period up to 72 h, which is considered the headache phase in migraine attacks, has never been reported so far. Our study has provided this missing information by focusing on the critical period of migraine attacks. From the technical viewpoint, it is difficult to judge whether increased signals in immunostaining reflect increased production or intracellular accumulation due to impaired release. Although RT-PCR is a gold standard method for quantifying mRNA expression levels, it cannot clarify the cell types responsible for mRNA synthesis. Importantly, our ISH data have elucidated the populations of TG neurons engaged in de novo *CGRP* mRNA synthesis. In TG tissue, CGRP is known to be expressed predominantly in small-diameter neurons with very thin unmyelinated nerve fibers [37,38,39,40], which is consistent with our findings. Meanwhile, the CGRP receptor components CLR and RAMP1 are expressed mainly in the larger neurons [39,40]. Consequently, under normal circumstances, there is little coexpression of CGRP with the CGRP receptor with a paracrine action of CGRP operative [41]. We found that CSD induced an upward shift in the *CGRP* mRNA-synthesizing TG neuron size. This finding raises the possibility that CSD may increase the occurrence of autocrine CGRP action, thus leading to higher exposure of the CGRP receptor to its ligand. Although the clinical implications of this novel finding remain elusive, we speculate that such conversion to the autocrine mechanism may contribute to migraine prolongation/recurrence and/or blunt the effectiveness of CGRP-targeted therapy.

There are several limitations to the present study. First, we used a CSD-based model, which is suitable for studying the disease state associated with migraine with aura. Hence, our findings may not be extrapolated to patients with migraine without aura. Moreover, we used untreated mice as controls, which cannot be regarded as real controls. Mice undergoing the experimental procedure except for KCl treatment might be better controls. However, the effect of KCl-induced nociceptive stimulation on the meningeal trigeminal afferents, which overlie the cerebral cortex, cannot be excluded from the CSD we adopted in the present study. As depicted in Figure 1A, scalp retraction was performed bilaterally. We did not observe a significant change in the cell size of *CGRP* mRNA-synthesizing TG neurons on the contralateral side. It is unlikely that the CSD-induced upward shift of the cell size of *CGRP* mRNA-synthesizing TG neurons on the ipsilateral side was due to a simple surgical effect. Second, we used only male mice to circumvent the potential effects of the menstrual cycle on the elicitability of CSD [42]. Because migraine is three times more common in women than in men [43], our data may not be applicable to female patients with migraine. Third, we analyzed TG neurons universally in our ISH experiment because it was hard for us to identify the exact trigeminal territory to which each of the examined TG neurons belonged. The innervation of trigeminal fibers to the cerebral vasculature is known to vary among the trigeminal subdivisions, with the ophthalmic division being responsible for the greatest vascular innervation [44]. Hence, if we had examined TG neurons in each subdivision individually, we could have obtained more significant data. Lastly, unlike RT-PCR, our ISH experiments were not able to provide information about the total production of *CGRP* mRNA. Moreover, because this is purely an in situ hybridization study, no data were available on the amount of CGRP at the extracellular or intracellular peptide level or CGRP release for reference.

Despite these limitations, the present study provides an important novel insight into the effect of CSD on the status of de novo *CGRP* mRNA synthesis in TG neurons. In particular, the CSD-induced shift in TG neuronal populations engaging in *CGRP* mRNA synthesis is a novel finding that may have relevance to the disease activity and therapeutic response, especially in cases of migraine with aura.

## 4. Materials and Methods

### 4.1. Animals

The present study was approved by the Laboratory Animal Care and Use Committee of Keio University (No. 14084). All experimental procedures were performed in accordance with the institution-approved protocols and EC Directive 86/609/EEC for animal experiments. Male C57BL/6 mice aged 8–10 weeks were purchased from CLEA Japan Inc. (Fujinomiya, Japan). A total of 15 mice (body weight: 25.1 ± 0.8 (mean ± SD)) were used for the present study. We failed to extract TG tissue after perfusion fixation in one animal. Hence, 14 out of the 15 mice were analyzed for the study. They were housed in ambient specific-pathogen-free conditions with a 12 h light/dark cycle and given food and water ad libitum.

### 4.2. CSD Induction

Under isoflurane anesthesia (1–2%), the mouse head was fixed in a stereotaxic apparatus. Systemic anesthesia was maintained using an anesthesia unit (model 410; Univentor Ltd., Zejtun, Malta). The rectal temperature was maintained at approximately 37 °C using a thermocontroller-regulated heating pad (BWT-100; Bioresearch Center Co., Ltd., Nagoya, Japan).

CSD was induced as described elsewhere [45]. Briefly, after a midline incision, the scalp was carefully reflected for skull exposure. Part of the exposed skull was thinned (0.5 mm in diameter) using a dental drill at 2 mm lateral and 4 mm posterior to the bregma on the left side (Figure 1A). CSD was induced in the left hemisphere by placing a cotton ball soaked with 1 M KCl solution over the thinned skull at 2 mm lateral and 4 mm posterior to the bregma. In all cases, CSD could be induced five times over 20–30 min (Figure 1B). After the occurrence of five CSD episodes was confirmed using a laser Doppler flowmeter (ALF 21, Advance Co., Ltd., Tokyo, Japan) set at 2 mm lateral and 2 mm posterior to the bregma on each side, the KCl-soaked cotton ball was removed, and the site was washed with isotonic NaCl. CSD was considered to be induced when a characteristic deflection appeared only on the side ipsilateral to KCl application in the laser Doppler flowmeter recording (Figure 1B). Continuous recordings of rCBF were stored on a multi-channel recorder (PowerLab 8/30; ADInstruments, Ltd., Sydney, Australia), and LabChart software version 8 (ADInstruments, Ltd.) was used for offline analysis. We confirmed that intraoperative hemodynamic parameters (heart rate (bpm) and systolic blood pressure (mmHg)) were within physiological ranges.

### 4.3. TG Tissue Excision

At 6 h, 24 h, 48 h, and 72 h after CSD induction, mice were euthanized with excess isoflurane. Thereafter, they were subjected to transcardial perfusion with saline followed by a fixative specially designed for in situ hybridization (ISH) (G-fix, Nippon Genetics Co., Ltd., Tokyo, Japan). Bilateral TG tissue was carefully excised from the skull. They were stored in distinct tubes containing the same fixative at 4 °C. TG tissue obtained from untreated mice served as control samples. The fixed tissue samples were embedded in paraffin on CT-Pro20 (Genostaff Co., Ltd., Tokyo, Japan) using G-Nox (Genostaff Co., Ltd., Tokyo, Japan) as a less toxic organic solvent for xylene and processed as 6 μm thick sections. Approximately 30 sections were prepared from each tissue sample.

### 4.4. Mouse TG ISH

ISH was performed with an ISH Reagent Kit (Genostaff Co., Ltd., Tokyo, Japan) according to the manufacturer’s instructions. Tissue sections were deparaffined with G-Nox (Genostaff Co., Ltd., Tokyo, Japan) and rehydrated through ethanol series and phosphate-buffered saline (PBS). The sections were fixed with 10% neutral buffered formalin (NBF) for 30 min at 37 °C and washed in distilled water, placed in 0.2% HCl for 10 min at 37 °C, washed in PBS, treated with 4 μg/mL Proteinase K (FUJIFILM Wako Pure Chemical Co., Osaka, Japan) in PBS for 10 min at 37 °C and washed in PBS, then placed within a Coplin jar containing 1× G-Wash (Genostaff Co., Ltd., Tokyo, Japan), equal to 1× SSC (3 M NaCl, 0.3 M sodium citrate). The targeted sequence for designing the mouse *CGRP* mRNA ISH probe was as follows:
CGRP: 5′-gaaaggctgatgaaagacacatatatttgcatccttcttagtattgaaaaacccttctccctttgacaggagctaaagctaagtgcagaataagttgcctattgtgcatcgtgttgtatgtgactctgtatccaataaacatgacagcatggttctggcttatctggtagcaaatatggtccccataaaccatcctgttgatgttgatgactctgctaaacctcaaggggatatgaaacactgcctcttgctcttctggggacacatggtaa-3′.


ISH for mouse *β-actin* mRNA was performed using MP-A-002 (Genostaff Co., Ltd., Tokyo, Japan). Hybridization was carried out with the probes (250 ng/mL) in G-Hybo-L (Genostaff Co., Ltd., Tokyo, Japan) for 16 h at 60 °C. After the hybridization, the sections were washed 3 times with 50% formamide in 2× G-Wash for 30 min at 50 °C and 5 times in TBST (0.1% Tween20 in TBS) at room temperature. After treatment with 1× G-Block (Genostaff Co., Ltd., Tokyo, Japan) for 15 min at room temperature, the sections were incubated with anti-DIG AP conjugate (Roche, Basel, Switzerland) diluted 1:2000 with G-Block (diluted 1/50) in TBST for 1 h at room temperature. The sections were washed twice in TBST and then incubated in 100 mM NaCl, 50 mM MgCl_2_, 0.1% Tween20, 100 mM Tris-HCl, pH 9.5. Visualization of mRNA detection was performed with NBT/BCIP Solution (Sigma, St. Louis, MO, USA). The sections were counterstained with Kernechtrot Stain Solution (Muto Pure Chemicals, Tokyo, Japan) and mounted with G-Mount (Genostaff Co., Ltd., Tokyo, Japan), then Malinol (Muto Pure Chemicals, Tokyo, Japan). Randomly selected TG sections obtained from each animal (*n* = 3 in control mice and *n* = 3 in CSD-subjected mice) were examined with a light microscope, and the numbers and cell diameters of TG neurons positive for *CGRP* mRNA were analyzed using NDP.view2 software (Hamamatsu Photonics, Hamamatsu, Japan) and Adobe Photoshop 2023 (San Jose, CA, USA) by an examiner blind to the identity of tissue sections. The cell size was measured when the nucleus was in the focal plane. In the analysis of neuronal diameters, we measured the longest diameters of neurons whose nuclei were visible on the section.

### 4.5. Statistical Analyses

All numerical data are expressed as means with SD or 95% confidence intervals (CI). The normality of numerical data distributions was assessed by the D’Agostino and Pearson normality test. Between-group comparisons for the *CGRP* mRNA-synthesizing TG neuronal density were performed using the Kruskal–Wallis test, followed by Dunn’s multiple comparison test. Between-group comparisons for cell diameters of *CGRP* mRNA-synthesizing TG neurons were conducted using one-way analysis of variance (ANOVA) followed by Dunnett’s multiple comparison test. Variance equality was evaluated by Bartlett’s test. In multiple comparisons, *p* values were adjusted for multiplicity. Between-group comparisons for cell diameters of *β-actin* mRNA-synthesizing TG neurons were conducted using Student’s *t*-test. Proportions of neuronal diameter distribution were evaluated using the chi-square test. *p* < 0.05 was considered statistically significant. Data were analyzed using GraphPad Prism 8 software (GraphPad Software, San Diego, CA, USA).

## Figures and Tables

**Figure 1 ijms-24-11578-f001:**
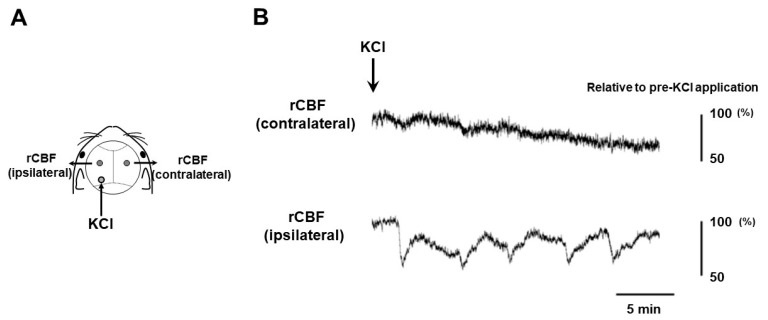
Experimental setting for CSD induction. (**A**) The sites of KCl application and laser Doppler flowmeter probe installation are illustrated. (**B**) Representative measures of regional cerebral blood flow (rCBF) over the contralateral and ipsilateral hemispheres to CSD induction are shown. rCBF was shown as a percentage point of the baseline value.

**Figure 2 ijms-24-11578-f002:**
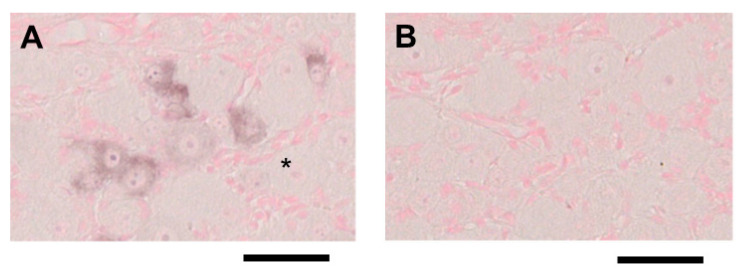
ISH for *CGRP* mRNA in TG tissue prepared from a control mouse. (**A**) *CGRP* mRNA expression was visualized by ISH with the Calca-4 anti-sense probe. *CGRP* mRNA was observed mainly in small- to medium-sized neurons. Large neurons, as indicated with the asterisk, were generally devoid of *CGRP* mRNA expression. The scale bar: 50 μm. (**B**) No significant staining was observed in ISH using the Calca-4 sense probe. The scale bar: 50 μm.

**Figure 3 ijms-24-11578-f003:**
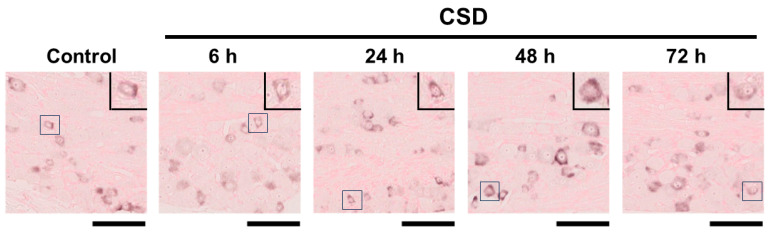
Representative *CGRP* mRNA in situ hybridization images of TG tissue in each experimental group. In each photograph, a high-powered image of a *CGRP* mRNA-synthesizing TG neuron is presented in the upper right corner. The highlighted TG neuron is shown in the quadrangle in each low-powered image. The bar below each low-powered image: 100 μm.

**Figure 4 ijms-24-11578-f004:**
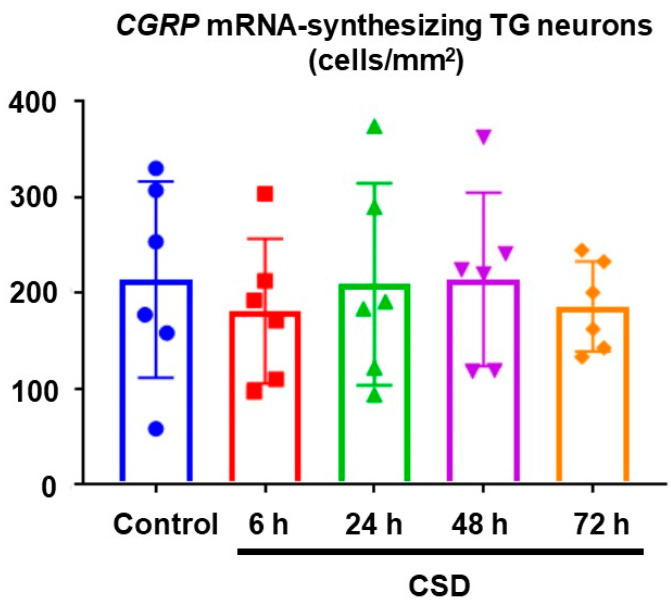
Density of *CGRP* mRNA-synthesizing TG neurons. The data are expressed as mean ± SD with scatter plots. Six sections were analyzed in each experimental group. Data were analyzed using the Kruskal–Wallis test and Dunn’s multiple comparison test.

**Figure 5 ijms-24-11578-f005:**
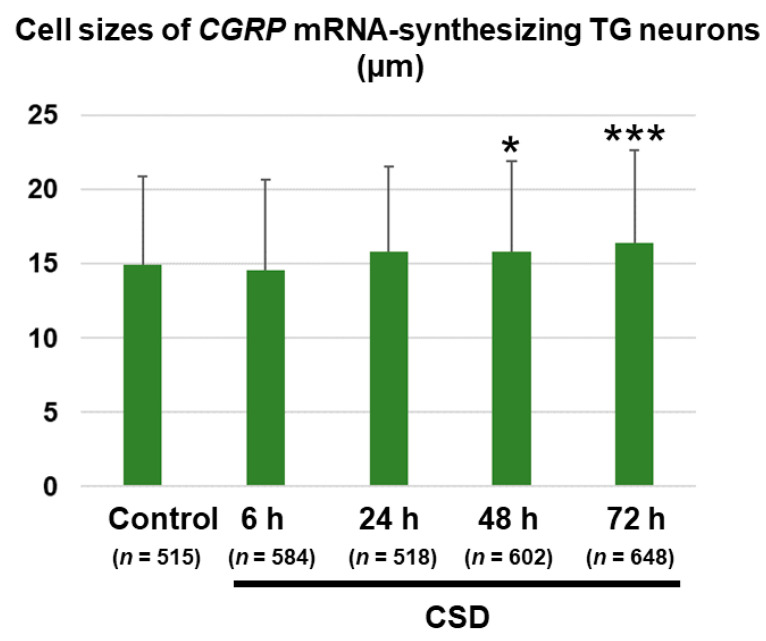
Cell size changes in *CGRP* mRNA-synthesizing TG neurons. All *CGRP* mRNA-synthesizing TG neurons were analyzed in six randomly selected tissue sections derived from each experimental group: control group, *n* = 515 (49–108 cells/section); 6 h post-CSD group, *n* = 584 (60–138 cells/section); 24 h post-CSD group, *n* = 518 (67–107 cells/section); 48 h post-CSD group, *n* = 602 (70–162 cells/section); 72 h post-CSD group, *n* = 648 (75–159 cells/section). Data are expressed as mean ± SD. Between-group comparison was performed using one-way ANOVA followed by Dunnett’s multiple comparison test. * *p* < 0.05 and *** *p* < 0.001 vs. Control.

**Figure 6 ijms-24-11578-f006:**
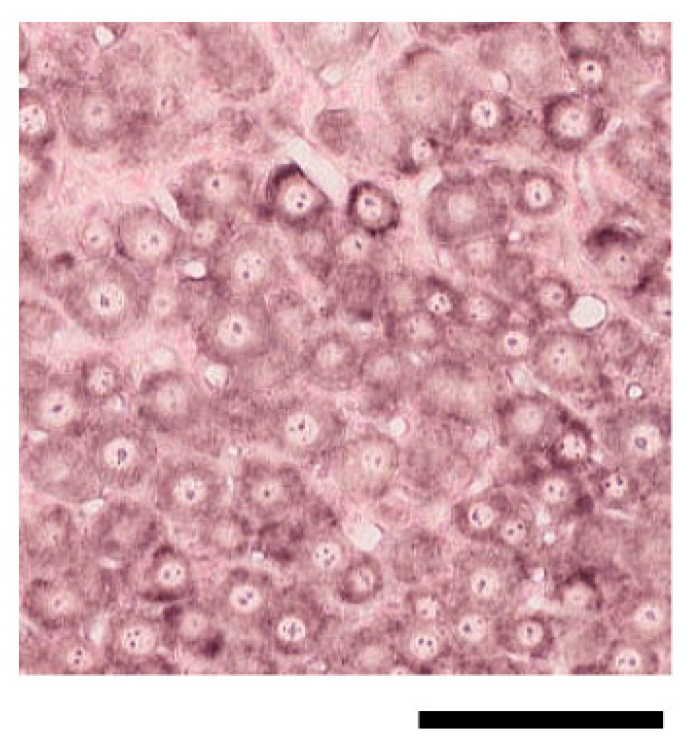
ISH for *β-actin* mRNA in TG tissue prepared from a control mouse. The scale bar: 100 μm.

**Figure 7 ijms-24-11578-f007:**
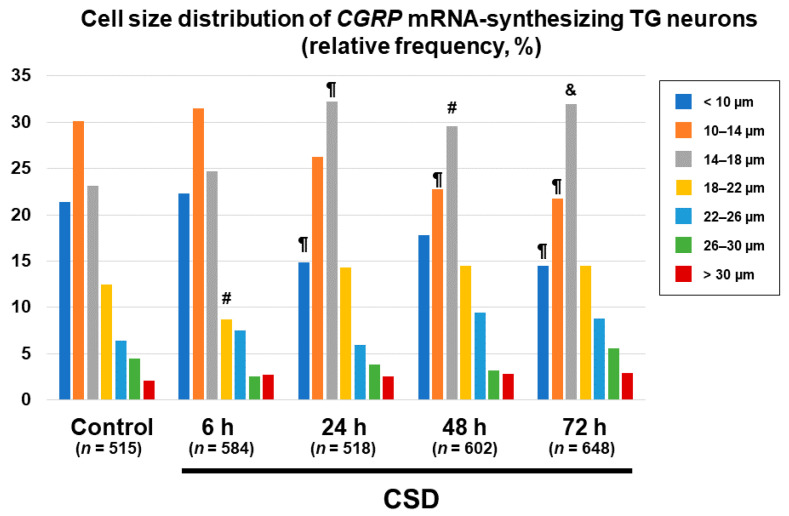
Changes in cell size distribution of *CGRP* mRNA-synthesizing TG neurons. Data are expressed as relative frequency in percentage of cell size fractions in each group: control group, *n* = 515 (49–108 cells/section); 6 h post-CSD group, *n* = 584 (60–138 cells/section); 24 h post-CSD group, *n* = 518 (67–107 cells/section); 48 h post-CSD group, *n* = 602 (70–162 cells/section); 72 h post-CSD group, *n* = 648 (75–159 cells/section). Cell size fractions were classified according to the right explanatory note. Between-group comparison was performed using the chi-square test. # *p* < 0.05, ¶ *p* < 0.001, and & *p*< 0.0001 vs. control in the cell size fraction.

## Data Availability

Relevant data generated and/or analyzed during this study are included in this published article.

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
