# Peer review of "Calcitonin Gene-Related Peptide mRNA Synthesis in Trigeminal Ganglion Neurons after Cortical Spreading Depolarization"

_ijms, 2023, doi:10.3390/ijms241411578_

Round 1

Reviewer 1 Report (Previous Reviewer 1)

I have a few comments that the authors need to address for revision.

1.  The control animals should be subjected to the same procedures as treatment groups except KCl, probably control solution here like aCSF. Untreated animals are not strict controls. Line 72. The authors mentioned ‘It is likely that the CSD-induced upward shift of the cell size of CGRP mRNA-synthesizing TG neurons on the ipsilateral side was due to a simple surgical effect.” Line 223-224. If the authors cannot exclude this possibility, then the whole manuscript won’t make any sense. I think the problem is controls were not set up strictly.

2. Figure 2 is not very clear. How did the authors define small, medium or large size of TG neurons?

3. CSD did not impact the number of CGRP mRNA-synthesizing TG neurons, but the neuron size. Unfortunately, in Figure 3, I failed to see the changes of neuron sizes.

4. Line 124-127, the authors need to show representative images of staining of β-actin in TG neurons.

5. How was these mice after CSD compared with the control animals? Any temporal behaviors (6, 24, 36, 48 and 72 hour) after CSD indicative of migraine? How does the size of CGRP mRNA-synthesizing TG neurons initiate migraine?

6. 5. The authors mentioned that CSD increased the CGRP-immunoreactive neurons in the literature. However, in the present studies CSD did not affect the number of CGRP-expressing cells, but the cellular sizes. What caused the discrepancy here?

7. Writing should be improved in the manuscript. For instance, Migraine adversely affects individuals and society as the second leading cause of years lived with disability worldwide (line 27); is it good to say no significant staining?

The language can be improved.

Author Response

We truly appreciate your insightful and encouraging comments.

  1. The control animals should be subjected to the same procedures as treatment groups except KCl, probably control solution here like aCSF. Untreated animals are not strict controls. Line 72. The authors mentioned ‘It is likely that the CSD-induced upward shift of the cell size of CGRP mRNA-synthesizing TG neurons on the ipsilateral side was due to a simple surgical effect.” Line 223-224. If the authors cannot exclude this possibility, then the whole manuscript won’t make any sense. I think the problem is controls were not set up strictly.

We understand the reviewer’s concern. Without the control proposed by the reviewer, we cannot exclude the traumatic effects derived from surgical interventions on the scalp and periosteal/bone tissue. However, even if we adopt such a control, we cannot rule out the effect of meningeal irritation by KCl solution in our CSD model. The most salient finding in the present study is the upward cell size change of CGRP mRNA-synthesizing TG neurons after CSD. We had prepared contralateral TG sections for a future research paper before the initial submission. Please note that scalp retraction was performed on the contralateral side for rCBF measurement (Figure 1A). Unlike the side ipsilateral to CSD induction, we did not detect a significant temporal change on the contralateral side (Lines 136–140). Although we acknowledge that we cannot wipe off the reviewer’s concern completely, the CSD-induced cell size change on the ipsilateral side is likely to be a specific finding.

  1. Figure 2 is not very clear. How did the authors define small, medium or large size of TG neurons?

As pointed out by the reviewer, the distinction among small, medium, and large TG neurons is not clarified. However, it is well-established that CGRP is expressed mainly in small-to-medium-sized neurons in TG tissue (J Mol Neurosci 2020;70:930-944, Nat Rev Neurol 2010;6:573-82.). In this figure, CGRP mRNA-synthesizing neurons are smaller than a representative large-sized neuron is indicated by asterisk. Moreover, it is shown that CGRP mRNA-synthesizing neurons were smaller than β-actin mRNA-synthesizing neurons, which represent the whole TG neurons (Lines 131-134).  

  1. CSD did not impact the number of CGRP mRNA-synthesizing TG neurons, but the neuron size. Unfortunately, in Figure 3, I failed to see the changes of neuron sizes.

Although we detected an upward shift in an upward shift of cell size of CGRP mRNA-synthesizing TG neurons, the differences were small (mean differences: 0.89 [95% CI: 0.00–1.77] μm, 48 h post-CSD group [n = 602] vs. control group [n = 515], P = 0.0492; 1.48 [95% CI: 0.61–2.35] μm, 72 h post-CSD group [n = 648] vs. control group [n = 515], P = 0.0001; Dunnett’s multiple comparison test) (Line 124–127). Hence, it is not surprising that the difference is not appreciable by simple inspection.

  1. Line 124-127, the authors need to show representative images of staining of β-actin in TG neurons.

Thank you very much for your suggestion. We have added a representative in situ hybridization image for β-actin mRNA as new Figure 6.  

  1. How was these mice after CSD compared with the control animals? Any temporal behaviors (6, 24, 36, 48 and 72 hour) after CSD indicative of migraine? How does the size of CGRP mRNA-synthesizing TG neurons initiate migraine?

The mice subjected to CSD exhibited trigeminal sensitization, photophobia, and hypoactivity. We reported these findings in our three previous papers (Sci Rep 2020;10:11408, Neurosci Res 2021;172:80-86, Int J Mol Sci 2022;23:13807.).

  1. The authors mentioned that CSD increased the CGRP-immunoreactive neurons in the literature. However, in the present studies CSD did not affect the number of CGRP-expressing cells, but the cellular sizes. What caused the discrepancy here?

To be honest, we do not understand what caused this discrepancy. When we failed to detect any significant change in the density of CGRP mRNA-synthesizing TG neurons, we got disappointed. We next explored the proportional distribution of cell size of CGRP mRNA-synthesizing TG neurons. As shown in Figure 7, we found an upward shift in the cell size of CGRP mRNA-synthesizing TG neurons. Considering the distinct distribution patterns of CGRP and its receptor, the upward shift in the cell size of CGRP mRNA-synthesizing TG neurons may increase the occurrence of autocrine CGRP action, thus leading to higher exposure of the CGRP receptor to its ligand. We think that this is a finding worth reporting, because this may affect the effectiveness of CGRP-targeted therapy (Lines 207–218).

  1. Writing should be improved in the manuscript. For instance, Migraine adversely affects individuals and society as the second leading cause of years lived with disability worldwide (line 27); is it good to say no significant staining?

This sentence is based on the findings in GBD2019 (Reference #1). Our manuscript underwent English editing prior to submission. However, this sentence was not corrected.   

We hope that you find our revised manuscript acceptable for publication.

Reviewer 2 Report (New Reviewer)

In the manuscript ijms-2499649, the Authors present data which, although preliminary, are very interesting and represent a good contribution to the research in the field of neurovascular disorders, migraine in particular.

The manuscript is sufficiently well organized and structured. Methodologies are rigorous and clearly stated. The discussion is organic.

This manuscript is appropriate for publication in the International Journal of Molecular Sciences. Therefore, I suggest accepting it after minor revision, including a re-reading to solve some typos in the text.

Minor revisions

1) For homogeneity, the bars indicating the scale in figure 1 should be described in the caption and not report the values, as shown in Figure 2.

2) In the cell size analyses, whether the statistical comparison was done on the number of neurons measured, the number of tissue sections, or the number of specimens is unclear. I hope on the latter.

3) In Figure 6, the error bars should be added.

4) I appreciate that the authors highlight the limitations of their research in the discussion. I would also add that the controls may not be “real controls”. In fact, CSD was not induced in the two mice, but if I understand correctly, they were not manipulated either. Is it possible that simple manipulation before the induction of CSD might have some effect? In my opinion, this cannot be ruled out with complete confidence.

Author Response

We truly appreciate your insightful and encouraging comments.

  1. For homogeneity, the bars indicating the scale in figure 1 should be described in the caption and not report the values, as shown in Figure 2.

We appreciate the comment. We have revised Figure 2 accordingly.

  1. In the cell size analyses, whether the statistical comparison was done on the number of neurons measured, the number of tissue sections, or the number of specimens is unclear. I hope on the latter.

The statistical comparison was done on the number of neurons measured. The method we used to make this comparison is described on Lines 322–329. We acknowledge that the numbers of animals used for the present study were not large. However, what we focused on here were the tissue density and size of TG neurons expressing CGRP mRNA. Eventually, we studied 500–600 TG neurons in each group, which was sufficient for statistical analysis.

  1. In Figure 6, the error bars should be added.

There are no error bars in Figure 6 (new Figure 7), because the figure shows frequency data for all TG neurons measured in each experimental group. Statistical analysis was performed by the chi-square test.

  1. I appreciate that the authors highlight the limitations of their research in the discussion. I would also add that the controls may not be “real controls”. In fact, CSD was not induced in the two mice, but if I understand correctly, they were not manipulated either. Is it possible that simple manipulation before the induction of CSD might have some effect? In my opinion, this cannot be ruled out with complete confidence.

In accordance with the reviewer’s comment, we have added a sentence in Discussion to acknowledge this (Lines 221–222). We received a similar comment from the other reviewer.

We hope that you find our revised manuscript acceptable for publication.

Reviewer 3 Report (New Reviewer)

This article provides a useful investigation as to the effect of cortical spreading depression on CGRP expression in trigeminal ganglia. The work performed, and the results themselves, are interesting. However there are a number of problems I have with the manuscript that prevent it being published in its current form.

Major:

It is very difficult to tell which TG sections were investigated in this work. We are told that sections were randomly chosen from mice, but it is not clear whether the authors made sure to score one ipsilateral and one contralateral section from each mouse. This comparison is central to the paper, and as such needs to be explained. More detail regarding this process is important.

Additionally, it appears the changes are being attributed to CSD, however there was no "sham" CSD mouse model (i.e. drilling without application of KCl). As such while these changes are likely from CSD, they could also arise due to the drilling procedure. This needs to be considered and discussed.

Further, were mice given pain relief post surgery? This is important to know, as there is a balance of animal welfare vs scientific conclusions that needs to be considered.

We also need information about which mouse was missed. I am assuming this was one of the control baseline mice, as this is n=2 in the paper. This number is very low, and if would be worth investigating another mouse to make up for this loss. I cannot find formal guidelines for independent replicates in IJMS, but I would be surprised if 2 was enough.

Lastly, the numbers do not seem to add up. Line 317 says 3 sections were examined from each animal, so we would expect N = 9, if there were three mice per condition (line 71). However figures report six sections per condition. This is contradictory, and needs to be addressed.

Minor:

Line 15: It is unclear whether "smaller neuronal populations" refers to the size of individual neurons, or the size of the population relative to the overlal population. Consider rephrasing.

Line 42: Some primary references here would be appreciated.

Figure 4, 5, and 6: The findings of the paper hinge on there being an increase on the ipsilateral side, but no change on contralateral side. Splitting these figures into these two sides (e.g. graph for ipsilateral vs contralateral) would therefore help support the findings. This distinction could also be improved in the main text, as it is often confusing as to whether the description is of total, ipsilateral only, or contralateral only.

Section 2.4: Is the first paragraph referring to total neurons, or just ipsilateral?

Line 131: "15..2" has one too many periods. Typo.

Figure 5: Figure title has a mixture of italic and standard text. Only CGRP should be italicised.

Figure 6: Is this the overall percentage? It would still be useful to get an idea of the variation in these numbers (e.g. refer to Rees et al 2022 "CGRP and the Calcitonin receptor are co-expressed in mouse, rat, and human TG neurons" Figure 3).

Lines 223 and 223: Please check this sentence, as the way it is written seems to contradict the findings of the study. 

Line 300: The sequence of the probe used in this study would be useful for future researchers.

Author Response

We appreciate your very kind and insightful comments.

  1. It is very difficult to tell which TG sections were investigated in this work. We are told that sections were randomly chosen from mice, but it is not clear whether the authors made sure to score one ipsilateral and one contralateral section from each mouse. This comparison is central to the paper, and as such needs to be explained. More detail regarding this process is important.

We apologize for the ambiguity. We excised bilateral TG tissue from each mouse. Of course, we treated the two samples from each animal distinctly, because the effect of CSD is crucial in the present study. All the samples were processed for in situ hybridization in the same manner. We have clarified this point by modifying the description in Materials and Methods (Lines 290–291).

  1. Additionally, it appears the changes are being attributed to CSD, however there was no "sham" CSD mouse model (i.e. drilling without application of KCl). As such while these changes are likely from CSD, they could also arise due to the drilling procedure. This needs to be considered and discussed.

We understand the reviewer’s concern. Without the control proposed by the reviewer, we cannot exclude the traumatic effects derived from the drilling. However, scalp retraction was made for rCBF measurement on the contralateral side as well (Figure 1A). Hence, the traumatic effect derived from scalp retraction was controlled in the present study. In the revised manuscript, we clearly acknowledge that the untreated mice did not serve as true controls (Lines 226–227).

  1. Further, were mice given pain relief post surgery? This is important to know, as there is a balance of animal welfare vs scientific conclusions that needs to be considered.

We did not give any pain relief medication to the mice. Although we previously reported that CSD-subjected mice exhibited trigeminal thermal hyperalgesia (Sci Rep 2020;10:11408, Neurosci Res 2021;172:80-86, Int J Mol Sci 2022;23:13807.), they did not seem to have spontaneous pain. Hence, we refrained from giving any medication. We thought that it was prudent to avoid unnecessary mediations, because they might have affected our experimental data.

  1. We also need information about which mouse was missed. I am assuming this was one of the control baseline mice, as this is n=2 in the paper. This number is very low, and if would be worth investigating another mouse to make up for this loss. I cannot find formal guidelines for independent replicates in IJMS, but I would be surprised if 2 was enough.

The reviewer is right. We apologize for the small number. However, we were not able to carry out additional experiment because of the lead author’s moving to another institution.

  1. Lastly, the numbers do not seem to add up. Line 317 says 3 sections were examined from each animal, so we would expect N = 9, if there were three mice per condition (line 71). However figures report six sections per condition. This is contradictory, and needs to be addressed.

Actually, we chose three sections only from the control samples. Hence, we examined six sections in all the experimental groups (Lines 328329).

  1. Line 15: It is unclear whether "smaller neuronal populations" refers to the size of individual neurons, or the size of the population relative to the overall population. Consider rephrasing.

We apologize for the ambiguity. We use smaller-sized instead of smaller in the revised manuscript.

  1. Line 42: Some primary references here would be appreciated.

We have added several primary references.

  1. Figure 4, 5, and 6: The findings of the paper hinge on there being an increase on the ipsilateral side, but no change on contralateral side. Splitting these figures into these two sides (e.g. graph for ipsilateral vs contralateral) would therefore help support the findings. This distinction could also be improved in the main text, as it is often confusing as to whether the description is of total, ipsilateral only, or contralateral only.

Thank you very much for your suggestion. The most interesting finding in the present study is the upward shift of CGRP mRNA-synthesizing TG neurons ipsilateral to CSD induction without the change in their total number. This should be highlighted. However, we obtained only negative results from the analyses of contralateral samples. Hence, we do not think that it is necessary to prepare figures for the contralateral data.  

  1. Section 2.4: Is the first paragraph referring to total neurons, or just ipsilateral?

Here, we refer to only ipsilateral data. To clarify, we have modified relevant sentences (Lines 122–130).

  1. Line 131: "15..2" has one too many periods. Typo.

We apologize for the error. We have corrected this problem.

  1. Figure 5: Figure title has a mixture of italic and standard text. Only CGRP should be italicised.

We apologize for the error. We have corrected this problem (new Figure 6).

  1. Figure 6: Is this the overall percentage? It would still be useful to get an idea of the variation in these numbers (e.g. refer to Rees et al 2022 "CGRP and the Calcitonin receptor are co-expressed in mouse, rat, and human TG neurons" Figure 3).

Thank you very much for kid suggestions. There are no error bars in Figure 6 (new Figure 7), because the figure shows frequency data for all TG neurons measured in each experimental group. Statistical analysis was performed by the chi-square test.

  1. Lines 223 and 223: Please check this sentence, as the way it is written seems to contradict the findings of the study.

We truly appreciate your kind comment. We were wrong. We have replaced likely with unlikely.

  1. Line 300: The sequence of the probe used in this study would be useful for future researchers.

Information on the sequence of the CGRP mRNA ISH probe can be found in Lines 311–314. 

We hope that you find our revised manuscript acceptable for publication.

This manuscript is a resubmission of an earlier submission. The following is a list of the peer review reports and author responses from that submission.

Round 1

Reviewer 1 Report

I have a few comments that the authors need to address for revision.

1.  The authors used untreated mice as controls (n=2), which was not strict controls. The control animals should be subjected to the same procedures as treatment groups except KCl, probably control solution here like CSF. N=2 is not enough for statistical analysis.

2. What is the unit of rCBF? The authors need to specify it in Figure 1B.

3. In Figure 3, why not counterstain with DAPI? Better images at all time points should be provided with higher magnification as well as resolution.

4. Did mice exhibit any behaviors after CSD indicating migraine-like state in humans?

5. The authors mentioned that CSD increased the CGRP-immunoreactive neurons in the literature. However, in the present studies CSD did not affect the number of CGRP-expressing cells, but the cellular sizes. What caused the discrepancy here? The authors need to discuss further.

6. Is it possible to measure CGRP local concentrations inside TG neurons after CSD? CSD did not significantly change the density of CGRP mRNA-synthesizing neurons in the ipsilateral TG. However, the cell sizes of CGRP mRNA-synthesizing TG neurons were significantly larger in the 48 h and 72 h post-CSD. It is very hard to convince the audience that neuron size, not other changes in TG, is linked with migraine induced by CSD.

Good.

Author Response

Dear Reviewer,

We were very glad to receive encouraging and constructive comments from you. In accordance with your comments, we have revised our original manuscript. Our point-by-point answers to your comments are as follows,

<Reviewer #1>

  1. The authors used untreated mice as controls (n=2), which was not strict controls. The control animals should be subjected to the same procedures as treatment groups except KCl, probably control solution here like CSF. N=2 is not enough for statistical analysis.

As pointed out by the reviewer, untreated mice were not considered strict controls in our study. Mice undergoing the experimental procedure except for KCl treatment seem to be better controls. However, the effect of KCl-induced nociceptive stimulation on the meningeal trigeminal afferents, which overlie the cerebral cortex, cannot be excluded in such animals. Unfortunately, this is a technical problem inherent to our CSD model, which does not require craniotomy. In the original manuscript, we raised this control issue as a limitation of our study. We have modified the text by adding the above explanation (lines: 212–216).

 As pointed out, we acknowledge that the numbers of animals used for the present study were not large, mainly for a financial reason. However, what we focused on here were the tissue density and size of TG neurons expressing CGRP mRNA. Eventually, we studied 500–600 TG neurons in each group, which was sufficient for statistical analysis.   

  1. What is the unit of rCBF? The authors need to specify it in Figure 1B.

We apologize for the unclarity. In Figure 1B, rCBF is indicated as the percentage point of the baseline rCBF measurable with Laser Doppler flowmetry. We have added this explanation to the figure legend (line: 79).

  1. In Figure 3, why not counterstain with DAPI? Better images at all time points should be provided with higher magnification as well as resolution.

Because we did not utilize fluorescent immunostaining, DAPI was not adequate for nuclear couterstaining. As you suggested, the number of cells can be better visualized by nuclear counterstaining. As stated in the Materials and Methods section, CGRP mRNA signal was visualized by alkaline phosphatase staining with NBT/BCIP salutation in the present study. In this case, nuclear couterstaining may obscure ISH signals. Hence, we refrained from using nuclear couterstaining.  

  1. Did mice exhibit any behaviors after CSD indicating migraine-like state in humans?

Yes, they did. We previously reported that our migraine model mice exhibited trigeminal sensitization, photophobia, and hypomotility, which are seen in actual patients with migraine (Tang C, et al. Sci Rep 2020;10:11408).

  1. The authors mentioned that CSD increased the CGRP-immunoreactive neurons in the literature. However, in the present studies CSD did not affect the number of CGRP-expressing cells, but the cellular sizes. What caused the discrepancy here? The authors need to discuss further.

We appreciate your suggestion. As stated in the Discussion section (lines 171–192), there have been a couple of studies reporting increased CGRP expression after CSD using immunostaining (reference numbers: 32 and 33). It should be borne in mind that immunostaining and ISH evaluate expression levels at different levels. Moreover, our timing of CGRP expression measurement was different from theirs. In addition, we emphasize that the present study is the first to explore the temporal profile of CGRP mRNA expression using ISH. We feel that our original manuscript provides sufficient information to address your concern.   

  1. Is it possible to measure CGRP local concentrations inside TG neurons after CSD? CSD did not significantly change the density of CGRP mRNA-synthesizing neurons in the ipsilateral TG. However, the cell sizes of CGRP mRNA-synthesizing TG neurons were significantly larger in the 48 h and 72 h post-CSD. It is very hard to convince the audience that neuron size, not other changes in TG, is linked with migraine induced by CSD.

Unfortunately, it is technically impossible to measure CGRP local concentrations inside TG neurons. In response to your suggestion, we have clarified this in the text (line 227–228).  

Although we were not able to address all the points raised by the reviewers for a limited duration for revisions, we hope that you find our revised manuscript eligible for publication. If you gave any inquiry about this manuscript, please do not hesitate to contact me. We look forward to hearing from you at the earliest opportunity.

Best regards,

Mamoru Shibata, M.D., Ph.D.

Professor, Department of Neurology, Tokyo Dental College Ichikawa General Hospital

5-11-13 Sugano, Ichikawa, Chiba 272-8513, Japan

Phone: +81-47-322-0151

Reviewer 2 Report

The authors have studied the expression of calcitonin gene-related peptide (CGRP) mRNA synthesis (by in situ hybridization) in the trigeminal ganglion (TG) neurons after cortical spreading depolarization (CSD). The authors reported that “CSD did not significantly change the density of CGRP mRNA-synthesizing neurons in the ipsilateral TG. However, the cell sizes of CGRP mRNA-synthesizing TG neurons were significantly larger in the 48 h and 72 h post-CSD”. Moreover, they say that “the CSD-induced upward cell size shift of CGRP mRNA-synthesizing TG neurons might be causative of greater disease activity and/or less responsiveness to CGRP-based therapy”.

Some points must be improved and clarified:

  1. The information showed in Figure 2 appears clearly in the text. Thus, this Figure could be removed.

  1. Figure 3. Photographs showing CGRP mRNA expression are very few, in fact, only one. The two photographs shown are from control animals. In addition, the quality of the photographs must be improved and high-magnifications of the photographs must be shown. The neuroanatomical information is very scarce; thus, authors must clearly show in a plate photographs from all the experimental groups studied (6h, 24h, 48h and 72h) in addition to the control group. This will serve to compare clearly the findings reported. Also show TG low-magnification photographs. This is a crucial point that must be improved. Photographs must clearly show the results indicated by the authors.

  1. Control group. Line 72: “we used untreated mice as controls (n = 2)” and line 62: “a relative noninvasive CSD induction method”. I understand that these control animals were not treated according to the method indicated in lines 241-244 (e.g., using dental grill). If this is the case, then an important control must be performed to confirm the results reported: animals with surgical procedure (midline incision, skull thinned with a dental drill) but without CSD induction.

  1. Line 100. Authors say: “Six tissue sections were studied in each experimental group”. I understand that from three animals (according to Figure 2) only six histological sections were studied per experimental group. Two sections per animal? Is this right? The number of sections is very low. This number must be increased. How many 6 µm sections were obtained from each TG?

  1. Lines 209-210. Authors say: “It was hard for us to identify the exact trigeminal territory to which each of the examined TG neurons belonged”. This could have been solved if authors would have performed a staining with cresyl violet. This must be performed.

  1. Lines 230-231. Authors say: “We failed in extracting TG tissue after perfusion fixation in one animal”. Why didn't the authors repeat it in another animal? I understand that this animal belonged to the control group.

  1. Indicate how the cell size was measured. Cell size was measured when the nucleus was in the focal plane?

  1. References. Check it according to the instructions for authors. For example, reference 2: 1866-1876; reference 4: 469-74.

  1. Lines 316-317. Supplementary material. I have not been able to obtain the files.

  1. The data reported are interesting but in some aspects this paper is preliminary, for example: 1) The experimental procedure must be also investigated in females; 2) The total amount of CGRP must be measured; 3) The study must be also performed in the contralateral side although the authors indicated in lines 42-43: “It is known that CSD causes sustained electrical activation of rat trigeminal ganglion (TG) neurons on the ipsilateral side [11]”; and 4) TG neurons contain also other peptides such as substance P and neurokinin A which are released like CGRP, What about these or other neuroactive substances after CSD induction?

In sum, I have four main methodological concerns: 1) Neuroanatomical findings must be clearly shown; 2) Another control experimental group (surgery without CSD induction) must be performed; 3) More TG sections must be studied; and 4) More experiments must be also carried out to increase the knowledge of the interesting topic studied.

Author Response

Dear Reviewer,

We were very glad to receive encouraging and constructive comments from you. In accordance with your comments, we have revised our original manuscript. Our point-by-point answers to your comments are as follows,

<Reviewer #2>

  1. The information showed in Figure 2 appears clearly in the text. Thus, this Figure could be removed.

In accordance with the reviewer’s advice, we have deleted the original Figure 2.

  1. Figure 3. Photographs showing CGRP mRNA expression are very few, in fact, only one. The two photographs shown are from control animals. In addition, the quality of the photographs must be improved and high-magnifications of the photographs must be shown. The neuroanatomical information is very scarce; thus, authors must clearly show in a plate photographs from all the experimental groups studied (6h, 24h, 48h and 72h) in addition to the control group. This will serve to compare clearly the findings reported. Also show TG low-magnification photographs. This is a crucial point that must be improved. Photographs must clearly show the results indicated by the authors.

In accordance with the reviewer’s suggestion, we have added a new figure displaying photographs from all the experimental groups studied (6h, 24h, 48h and 72h) in addition to the control group (Figure 3).

  1. Control group. Line 72: “we used untreated mice as controls (n = 2)” and line 62: “a relative noninvasive CSD induction method”. I understand that these control animals were not treated according to the method indicated in lines 241-244 (e.g., using dental grill). If this is the case, then an important control must be performed to confirm the results reported: animals with surgical procedure (midline incision, skull thinned with a dental drill) but without CSD induction.

The same concern was raised by the other reviewer. We agree that the control mice that you suggested are better than untreated ones. However, the effect of KCl-induced nociceptive stimulation on the meningeal trigeminal afferents, which overlie the cerebral cortex, cannot be excluded in such animals. Unfortunately, this is a technical problem inherent to our CSD model, which does not require craniotomy. In the original manuscript, we raised this control issue as a limitation of our study. We have modified the text by adding the above explanation (lines: 212–216).

  1. Line 100. Authors say: “Six tissue sections were studied in each experimental group”. I understand that from three animals (according to Figure 2) only six histological sections were studied per experimental group. Two sections per animal? Is this right? The number of sections is very low. This number must be increased. How many 6 µm sections were obtained from each TG?

As pointed out, we acknowledge that the numbers of animals used for the present study were not large, mainly for a financial reason. This undermines the statistical power of our study. However, what we focused on here were the tissue density and size of TG neurons expressing CGRP mRNA. Eventually, we studied 500–600 TG neurons in each group, which was sufficient for statistical analysis.   

  1. Lines 209-210. Authors say: “It was hard for us to identify the exact trigeminal territory to which each of the examined TG neurons belonged”. This could have been solved if authors would have performed a staining with cresyl violet. This must be performed.

As pointed out, we should have performed cresyl violet staining instead of kernechtrot staining. To be honest, we sometimes found it difficult to determine which trigeminal territory (ophthalmic, maxillary, or mandibular territory) the examined TG neurons belonged to. Initially, we tried to individually examine TG neurons in the three territories. It was too hard to accomplish. That is why we raised this point as a limitation of the present study.   

  1. Lines 230-231. Authors say: “We failed in extracting TG tissue after perfusion fixation in one animal”. Why didn't the authors repeat it in another animal? I understand that this animal belonged to the control group.

We have to acknowledge that this is a big weakness of the present study. As stated above, we were not able to repeat for a financial reason.

  1. Indicate how the cell size was measured. Cell size was measured when the nucleus was in the focal plane?

As stated in the Materials and Methods section (line: 309), we measured cell size employing the NDP.view2 software (Hamamatsu Photonics, Hamamatsu, Japan). As you pointed out, cell size was measured when the nucleus was in the focal plane. We have added this sentence to the text (lines: 311-312). 

  1. Check it according to the instructions for authors. For example, reference 2: 1866-1876; reference 4: 469-74.

We apologize for the mess. Now, all the references are correctly styled.

  1. Lines 316-317. Supplementary material. I have not been able to obtain the files.

We are sorry. This was a pure mistake, Actually, there are no supplementary material.

  1. The data reported are interesting but in some aspects this paper is preliminary, for example: 1) The experimental procedure must be also investigated in females; 2) The total amount of CGRP must be measured; 3) The study must be also performed in the contralateral side although the authors indicated in lines 42-43: “It is known that CSD causes sustained electrical activation of rat trigeminal ganglion (TG) neurons on the ipsilateral side [11]”; and 4) TG neurons contain also other peptides such as substance P and neurokinin A which are released like CGRP, What about these or other neuroactive substances after CSD induction?

We appreciate your excellent suggestions. We will address the points that you raised in future studies.

Although we were not able to address all the points raised by the reviewers for a limited duration for revisions, we hope that you find our revised manuscript eligible for publication. If you gave any inquiry about this manuscript, please do not hesitate to contact me. We look forward to hearing from you at the earliest opportunity.

Best regards,

Mamoru Shibata, M.D., Ph.D.

Professor, Department of Neurology, Tokyo Dental College Ichikawa General Hospital

5-11-13 Sugano, Ichikawa, Chiba 272-8513, Japan

Phone: +81-47-322-0151

Fax: +81-47-325-0046

[email protected]/[email protected]

Round 2

Reviewer 1 Report

No.

The authors addressed most of my comments, and admitted the flaws of the manuscript that they cannot do extra experiments.

Reviewer 2 Report

The authors have improved the Ms. However, some important information is lacking.

  1. The information showed in Figure 2 appears clearly in the text. Thus, this Figure could be removed. This has been done by the authors. 
  1. Figure 3. Photographs showing CGRP mRNA expression are very few, in fact, only one. The two photographs shown are from control animals. In addition, the quality of the photographs must be improved and high-magnifications of the photographs must be shown. The neuroanatomical information is very scarce; thus, authors must clearly show in a plate photographs from all the experimental groups studied (6h, 24h, 48h and 72h) in addition to the control group. This will serve to compare clearly the findings reported. Also show low-magnification photographs in which the TG can be observed. This is a crucial point that must be improved. Photographs must clearly show the results indicated by the authors. This has been improved.
  1. Control group. Line 72: “we used untreated mice as controls (n = 2)” and line 62: “a relative noninvasive CSD induction method”. I understand that these control animals were not treated according to the method indicated in lines 241-244 (e.g., using dental grill). If this is the case, then an important control must be performed to confirm the results reported: animals with surgical procedure (midline incision, skull thinned with a dental drill) but without CSD induction. This control is missing. It must be performed.
  1. Line 100. Authors say: “Six tissue sections were studied in each experimental group”. I understand that from three animals (according to Figure 2) only six sections were studied per experimental group. Two sections per animal? Is this right? The number of sections is very low. This number must be increased. How many 6 µm sections were obtained from each TG? Both questions have not been answered by the authors. In the case that only two sections/animal were studied (six histological sections/experimental group), it is important to know the total number of sections obtained from each TG.
  1. Lines 209-210. Authors say: “It was hard for us to identify the exact trigeminal territory to which each of the examined TG neurons belonged”. This could have been solved if authors would have performed a staining with cresyl violet. This must be performed. The response indicated by the authors is OK.
  1. Lines 230-231. Authors say: “We failed in extracting TG tissue after perfusion fixation in one animal”. Why didn't the authors repeat it in another animal? I understand that this animal belonged to the control group. Although I fully understand the authors, the lack of funding is not an excuse for properly conducting experiments.
  1. Indicate how the cell size was measured. Cell size was measured when the nucleus was in the focal plane? The response indicated by the authors is OK.
  1. References. Check it according to the instructions for authors. For example, reference 2: 1866-1876; reference 4: 469-74. The response indicated by the authors is OK.
  1. Lines 316-317. Supplementary material. I have not been able to obtain the files. The response indicated by the authors is OK.
  1. The data reported are interesting but in some aspects this paper is preliminary, for example: 1) The experimental procedure must be also investigated in females; 2) The total amount of CGRP must be measured; 3) The study must be also performed in the contralateral side although the authors indicated in lines 42-43: “It is known that CSD causes sustained electrical activation of rat trigeminal ganglion (TG) neurons on the ipsilateral side [11]”; and 4) TG neurons contain also other peptides such as substance P and neurokinin A which are released like CGRP, What about these or other neuroactive substances after CSD induction? Authors say that these points will be raised in future studies. 

In sum, some important concerns indicated in the first round have not been answered: 1) Control experimental group (midline incision + thinned using a dental grill and without CSD induction); 2) More TG sections must be studied; and 3) More experiments must be also carried out to increase the knowledge on the topic studied.